# Engineered EVs for Oxidative Stress Protection

**DOI:** 10.3390/ph14080703

**Published:** 2021-07-21

**Authors:** Anna Maria Tolomeo, Santina Quarta, Alessandra Biasiolo, Mariagrazia Ruvoletto, Michela Pozzobon, Giada De Lazzari, Ricardo Malvicini, Cristian Turato, Giorgio Arrigoni, Patrizia Pontisso, Maurizio Muraca

**Affiliations:** 1Department of Women’s and Children’s Health, University of Padova, Via Giustiniani, 35128 Padova, Italy; annamaria.tolomeo@unipd.it (A.M.T.); michela.pozzobon@unipd.it (M.P.); giada.delazzari@phd.unipd.it (G.D.L.); rmalvicini@favaloro.edu.ar (R.M.); muraca@unipd.it (M.M.); 2L.i.f.e.L.a.b. Program, Consorzio per la Ricerca Sanitaria (CORIS), Veneto Region, Via Giustiniani, 35128 Padova, Italy; alessandra.biasiolo@unipd.it (A.B.); mariagrazia.ruvoletto@unipd.it (M.R.); patrizia@unipd.it (P.P.); 3Department of Medicine, University of Padova, Via Giustiniani, 35128 Padova, Italy; 4Institute of Pediatric Research “Città della Speranza”, Corso Stati Uniti, 35127 Padova, Italy; 5Instituto de Medicina Traslacional, Trasplante y Bioingeniería (IMeTTyB-CONICET), Universidad Favaloro-CONICET, Solís 453, Buenos Aires C1078AA, Argentina; 6Department of Molecular Medicine, University of Pavia, 9 Via A Ferrata, 27100 Pavia, Italy; cristian.turato@unipv.it; 7Department of Biomedical Sciences, University of Padova, 35121 Padova, Italy; giorgio.arrigoni@unipd.it; 8Proteomic Center of Padova, University of Padova, 35131 Padova, Italy

**Keywords:** extracellular vesicles, SerpinB3, oxidative stress, cytoprotection

## Abstract

Extracellular vesicles (EVs) are increasingly studied as vectors for drug delivery because they can transfer a variety of molecules across biological barriers. SerpinB3 is a serine protease inhibitor that has shown a protective anti-apoptotic function in a variety of stressful conditions. The aim of this study was to evaluate protection from oxidative stress-induced damage, using extracellular vesicles that overexpress SerpinB3 (EVs-SB3) in order to enhance the effect of extracellular vesicles on cellular homeostasis. EVs-SB3s were obtained from HepG2 cells engineered to overexpress SerpinB3 and they revealed significant proteomic changes, mostly characterized by a reduced expression of other proteins compared with EVs from non-engineered cells. These EV preparations showed a significantly higher protection from H_2_O_2_ induced oxidative stress in both the hepatoma cell line and in primary cardiomyocytes, compared to cells treated with naïve EVs or SerpinB3 alone, used at the same concentration. In conclusion, the induction of SerpinB3 transgene expression results in the secretion of EVs enriched with the protein product that exhibits enhanced cytoprotective activity, compared with naïve EVs or the nude SerpinB3 protein.

## 1. Introduction

In the past decade, extracellular vesicles (EVs) have been recognized as potent vehicles of intercellular communication, due to their capacity to transfer proteins, lipids, and nucleic acids, thereby influencing various physiological and pathological processes in target cells. Because of their ability to function as an intercellular cargo transfer system, EVs have been used as vehicles for delivering synthetic drugs as well as therapeutic proteins and miRNAs [1].

Serpins (serine protease inhibitors) are inhibitors of both serine and cysteine proteases [2], and are characterized by a marked conformational flexibility, which allows proteolysis in different biological processes such as inflammation, blood coagulation and pressure regulation, condensation of the chromatin, protein folding, and tumor progression to be controlled [3]. They can also act independently of their protease inhibitory functions, e.g., as chaperones or hormonal transporters [4]. In particular, SerpinB3 (SB3) can be physiologically detected in the superficial and intermediate layers of the normal squamous epithelium but it is also highly expressed in the hepatic stem / progenitor cell compartment that is composed of quiescent cells that proliferate under conditions of oxidative stress [5]. This molecule has shown a protective anti-apoptotic function, unrelated to its proteinase inhibition activity, in a variety of stressful conditions [6]. In particular, after radiation exposure it was able to determine an inhibitory effect of MAP family kinase JNK [7] or p38 [8]. More recently, SB3 was found to be localized in the internal mitochondrial compartments, where its binding to the respiratory complex determined cell protection from the toxicity of pro-oxidizing chemotherapeutic agents [9] by reducing the generation of ROS, a crucial step for opening the transition pore of mitochondrial permeability, leading to cell death by apoptosis [10].

On the basis of the above considerations, the aim of this study was to evaluate protection by oxidative stress-induced damage, using extracellular vesicles that overexpress SerpinB3 in order to enhance the already positive effect of extracellular vesicles on cellular homeostasis.

## 2. Results

### 2.1. EV-SerpinB3 Production and Characterization

The initial phase of the experiments aimed to optimize the conditions to obtain highly efficient EV production in terms of number and biological activity. Before each EV isolation step, every batch of cells used for the production of EVs was characterized by a real-time PCR analysis and by immunofluorescence for SB3 expression, as reported in the example of Figure 1.

A tunable resistive pulse sensing (TRPS) analysis of the EVs isolated from HepG2 and HepG2/SB3 demonstrated two homogeneous populations with particle sizes mostly below 100 nm, with no significant differences between naïve EVs (nEVs) and HepG2/SB3-derived EVs (EVs-SB3) (Figure 2A,B). Notably, the EV particles/cell increased by approximately two-fold in the medium of SerpinB3-conditioned HepG2s (Figure 2C). Transmission electron microscopy (TEM) identified the EVs as a group of heterogeneous spheroids with sizes ranging from 30 to 300 nm (Figure 2D,E). The RPS and TEM analysis confirmed therefore that almost all the nanoparticles isolated in the present work can be referred to as “small EVs”, according to the MISEV2018 criteria [11]. No apoptotic bodies were detected. SerpinB3 was detected in EVs-SB3, while only trivial levels were detected in nEVs, as demonstrated by the Elisa and Western blot results and these results confirmed an efficient transfection (Figure 2F,G). The EV surface of both nEVs and EVs-SB3 was analyzed and the classical tetraspanin (CD63, CD9), the adhesion molecules such as CD29 and the angiogenic proteins CD146 were identified as the characteristic markers of these lipid-bilayer particles (Figure 2H). While the absence of HLA-DR underlines their high biocompatibility, this also suggests a low rejection rate expectancy.

### 2.2. Proteomic Analysis of EV

The protein composition of EVs, both from nEVs and from EVs-SB3 preparations, was evaluated with a label-free MS-based proteomic approach. Appendix A lists all proteins identified with high confidence and all the relevant parameters used to assess the reliability of protein identification and quantification. The proteomic analysis led to the reliable identification of more than 250 different protein groups. These were subjected to a categorization to highlight enriched the Gene Ontology (GO) terms using the bioinformatic tools gProfiler and Revigo (Appendix A). As reported in Figure 3A, most of the enriched GO terms are strongly related to vesicle-mediated transport, secretion, response to stress, immune system process, regulation of peptidase activity, and proteolysis, with the latter ones reflecting the numerous Serpin proteins identified in the EV samples. 

Proteins that were differently abundant in the nEVs and in the EVs-SB3 samples are reported in Table 1, together with their p value and Fold Change.

A STRING network analysis of these proteins (Figure 3B) highlights that most of them are physically/functionally connected and many are involved in vesicle mediated transport (GO:0016192; FDR 1.4 × 10^−4^ and GO:0031982; FDR 2.37 × 10^−5^) and endowed with catalytic activity (GO:0003824, FDR 6.74 × 10^−5^). To confirm the reliability of these results, a Western Blot analysis was also carried out for one more abundant protein (haptoglobin) and one less abundant protein (Appendix A). By looking at the 33 proteins listed in Table 1, it is immediately obvious that most of them show a negative FC and are therefore less abundant in EVs-SB3, compared to the nEVs preparations. Only one protein, namely, haptoglobin (HP), showed an increased abundance in EVs-SB3. It is worth noting that haptoglobin is an acute phase protein that, besides binding to haemoglobin, also acts as an antioxidant and antimicrobial agent, playing a relevant role in the host defence responses to infection and inflammation [12].

#### EV Uptake Detection

In order to verify whether the EVs produced were internalized by the cells, EVs were labeled with a Vybrant™ DiI Cell-Labeling Solution. The signal of red fluorescence was detectable as early as 6 h after treatment and significantly increased at 24 h. These results indicate an efficient internalization of both extracellular vesicles within target cells. No quantitative differences were observed between the uptake of nEVs compared to EVs-SB3 (Figure 4).

### 2.3. Biological Activity of EV-SerpinB3 In Vitro

Different EV preparations were analyzed for their ability to protect from oxidative damage. As an experimental in vitro model, we used H_2_O_2_-induced oxidative stress. In order to detect the optimal oxidative stress conditions, HepG2 control cells were seeded in 96 well plates and treated with increasing amounts of H_2_O_2_ (50 μM–1 mM) for 72 h and then the MTT assay was carried out to determine cell viability. The concentration of 250 μM H_2_O_2_ was identified as the best experimental condition, since it induced 45–55% of cell mortality and was considered optimal to study the potential protective activity by EV preparations as well as by the recombinant human SB3, used as a positive control.

The second step aimed to identify the concentration range of EVs able to induce a protective effect in the in vitro model of oxidative damage. A wide concentration range was tested, ranging from 10^9^ to 10^−1^ EVs/mL and a dose-dependent protective effect was observed. The optimal EV-SB3 concentration to achieve a protective effect in vitro was equal to 10^7^ EVs/mL. This positive result was higher than that determined by recombinant SB3 alone, used at the same SB3 concentration (109 ng/mL) detected by ELISA in the EVs-SB3 preparation used in this experiment (Figure 5A).

These encouraging results were also confirmed in the primary cell line of cardiomyocytes. The protective effect of EV-SB3 was significantly higher than that of the nEVs and of the recombinant SerpinB3 alone after treating the cells with 250 μM of H_2_O_2_ for 2 h (Figure 5B). These results were also confirmed by the live and dead assay (Figure 5C). These findings provide the basis for the use of EVs functionalized with SerpinB3 for cytoprotection from oxidative stress damage.

## 3. Material and Methods

### 3.1. Characterization of SerpinB3 Expression in Transfected HepG2 Cells

Hepatoma cells (HepG2 cell line) (ATCC, Manassas, VA, USA) were engineered to overexpress SB3 by transfection with a plasmid expression vector containing the human SB3 gene (pCDNA3 / SB3), or with the plasmid vector alone (pcDNA3.1D/V5-His-TOPOTM) as the control, using Lipofectamine Reagent Plus as the transfecting agent according to the recommended indications (Invitrogen, Carlsbad, CA, USA).

Transfected cells were selected using G418 (Geneticin) (Sigma-Aldrich, St. Louis, MO, USA) as a selective compound, and were added to the culture medium at every change. The clonal selection of transfected cells was conducted by serial dilutions of trypsinized cells that were subsequently seeded in 96 well plates, 48 h after transfection. When cellular colonies were visible, they were picked up and expanded into 12 well plates. Cells at 80% confluence were then trypsinized and tested for SB3 expression by a real-time PCR. Only clones expressing high levels of mRNA SB3 were expanded and analysed for protein expression by immunofluorescence. The clonal expansion of HepG2 cells transfected with the plasmid vector alone (controls) were carried out in presence of the medium with G418 as the selective compound. The HepG2 cell clones were used to obtain EVs-SB3 and their relative controls (nEVs) for the present project. 

#### 3.1.1. Immunofluorescence

The expression of the SB3 protein in transfected clones was assessed by immunofluorescence. Cells were seeded on slides (5 × 10^3^ cells/well in a 12 well plate) and cultured overnight. Cells were then fixed in 4% paraformaldehyde, permeabilized with 0.2% Tryton X 100, and blocked with 5% Goat serum in 1% BSA in PBS. Slides were then incubated with 8 μg/mL of a rabbit polyclonal antibody against SB3 (Hepa-Ab, Xeptagen, Marghera, VE, Italy), washed with 0.1% Tween 20 in PBS, and incubated with 488 AlexaFluor anti-rabbit secondary antibody (1:500 dilution) (Invitrogen, Carlsbad, CA, USA).

Cellular nuclei were counter stained with DAPI (Sigma-Aldrich, St. Louis, MO, USA). Slides were mounted with ELVANOL (Sigma-Aldrich, St. Louis, MO, USA) and observed under a fluorescence microscope utilizing the optical sectioning of Apotome.2 (Axiovert 200M, Carl Zeiss MicroImaging GmbH, Göttingen, Germany) for high quality images.

#### 3.1.2. Quantitative Real-Time RT-PCR

The levels of SB3 mRNA were measured in cultured cells by a real-time PCR using the SYBR® green method. The total RNA was extracted using RNasy Trizol (Invitrogen, Carlsbad, CA, USA) according to the manufacturer’s instructions. After determination of the purity and the integrity of the total RNA, complementary DNA synthesis and quantitative real-time PCR reactions (RT-PCR) were carried out as previously described [9] using the CFX96 Real-Time instrument (Bio-Rad Laboratories Inc, Hercules, CA, USA). The relative expression was generated for each sample by calculating 2-ΔCt.

The designed oligonucleotide sequences of the SB3 primers were the following: sense 5’-GCA AAT GCT CCA GAA GAA AG-3’ and reverse 5’-CGA GGC AAA ATGAAAA AGA TG-3’. Glyceraldehyde-3-phosphate dehydrogenase (GAPDH) was used as the internal reference and was co-amplified with target samples using identical real-time PCR conditions.

### 3.2. Extracellular Vesicle Production and Characterization

The transfected cells were amplified up to 4–6 × 10^8^ for both transfected cell lines, HepG2/SB3 and HepG2/CTR, for each batch of EV production. The cells were grown up to 80% confluence. The day before the isolation of the EVs, the cells were switched to a serum-free medium. The culture medium of all the flasks was used for the isolation of the EVs by ultrafiltration. 

The culture medium was centrifuged at 1200 rpm for 6 min to discard dead cells and debris and filtered through a 0.22 nm filter (filter unit syringe driven, Millex-GP, Merck Millipore, Darmstadt, Germany). The supernatant was loaded onto an Amicon filter device (Amicon filters Ultra-15, regenerate cellulose 100,000 NMWL; Merck Millipore, Darmstadt, Germany), centrifuged at 3200 g at 4 °C for 15 min, and the concentrate was then collected.

Particle concentration and size distribution were analyzed by tunable resistive pulse sensing (TRPS) technology with the qNano instrument (Izon Science, Christchurch, New Zealand), using a NP150 membrane. The concentration of particles was standardized using a CPC100 calibration solution diluted 1:10,000 (110 nm mean carboxylate polystyrene beads; raw concentration 1.00 × 10^12^).

For surface biomarker characterization, the MACSPlex exosome kit (Miltenyi Biotech) was used, following the manufacturer’s instructions.

#### 3.2.1. Transmission Electron Microscopy

One drop of EVs solution (about 25 µL) was placed on a 400-mesh holey film grid; after staining with 1% uranyl acetate (for 2 min), the sample was observed with a Tecnai G2 (FEI) transmission electron microscope operating at 100 kV. Images were captured with a Veleta (Olympus Soft Imaging System) digital camera.

#### 3.2.2. SerpinB3 Quantification by ELISA

The protein concentration of EV preparations was quantified by a bicinchoninic acid assay (BCA) protein kit (PIERCE MERCK, Darmstadt, Germany), as directed by the manufacturer, using BSA as a standard on a Victor X3 microplate reader (Perkin Elmer, Waltham, MA, USA).

To assess the presence of SB3, 20 μL of EVs were diluted 1:2 with a PBS buffer with a pH of 7.4 and kept on ice at 4 °C for 15 min. After a step at −80 °C for 5 min, the samples were vortexed for 10 s and placed in ice for 15 min before use. SB3 concentration was measured by a sandwich ELISA (HEPA Lisa kit, Xeptagen, Venice, Italy), following the manufacturer’s instructions. Briefly, 100 μL of 1:80 diluted EVs in a PBS with a pH of 7.4 were incubated for 1 h at room temperature on plates coated with rabbit anti-human SB3 capture Ab (10 μg/mL in PBS, pH 7.4). A standard curve, obtained by dilution of the recombinant SB3 from 16 to 0.25 ng/mL, was also included. All samples were tested in duplicate. After washing, SB3 was revealed by incubation with 100 μL HRP-conjugated streptavidin secondary anti-SB3 Ab (0.5 μg/mL). The plate was developed with a ready-to-use 3,3′, 5,5′-tetramethylbenzidine (TMB) substrate solution. The reaction was stopped with 1 mol/L HCl (100 μL) and absorbance at 450 nm was measured on a microplate reader (Victor X3; Perkin Elmer, Waltham, MA, USA).

### 3.3. Proteomic Analysis of EVs

EVs from the control and the SB3 expressing cells were characterized with a mass spectrometry (MS)-based proteomic approach.

A total of 50 μg of proteins from CTR and SB3 EVs (three independent preparations) were reduced with dithiothreitol (Fluka, Buchs, Switzerland), alkylated with iodoacetamide (Sigma-Aldrich, St. Louis, MO, USA), and digested with sequencing grade modified trypsin (Promega, Madison, WI, USA) using a filter aided sample preparation protocol (FASP), as previously detailed [13].

A label-free LC-MS/MS analysis was performed with an LTQ-Orbitrap XL mass spectrometer (Thermo Fisher Scientific, Waltham, MA, USA) interfaced with a nano-HPLC Ultimate 3000 (Dionex–Thermo Fisher Scientific, Waltham, MA, USA). Samples were loaded into a trap column (NanoEase Symmetry300, C18, 5 µm, Waters) at a flow rate of 8 µL/min and tryptic peptides were separated at a flow rate of 250 nL/min in a pico-frit capillary column (11 cm, 75 µm internal ID, 15 µm tip, New Objective) packed in-house with C18 material (ReproSil, 300Å, 3 μm; Dr. Maisch HPLC GmbH). A linear gradient of ACN/0.1% formic acid (FA) from 3 to 50% was applied in 90 min. The instrument operated in a data-dependent mode alternating a full scan in the Orbitrap (60,000 nominal resolution) in the 300–1700 m/z range with lower resolution MS/MS scans of the 10 most intense ions acquired in the linear ion trap. To reduce possible carry-over, a blank sample was injected after each EV preparation. 

Raw files were analyzed with the MaxQuant/Andromeda search engine software package (v.1.5.1.2). Proteins were searched against the Human section of the UniProt database (version July 2018, 95,057 entries). Trypsin was set as the enzyme with one missed cleavage allowed. Precursor and fragment tolerance were set to 10 ppm and 0.6 Da, respectively. The carbamidomethylation of cysteine residues was set as a fixed modification while methionine oxidation was set as a variable modification. Results were filtered to keep into account only the proteins identified with at least two peptides and with a false discovery rate (FDR) of 0.01, both at the peptide and protein level. The intensity values given by the software were used to assess differences in the composition of nEVs and the EVs-SB3 samples. Proteins were considered to have a significantly different abundance with *p* < 0.05 (*t*-test) and a fold change (FC) ≥2 or ≤−2.

Bioinformatic analyses were conducted using the String network software [14], g-Profiler [15] and Revigo to reduce and visualize the Gene Ontology (GO) enriched terms [16].

### 3.4. Western Blot Analysis of Selected EV Proteins

Extracellular vesicles lysates were fractionated by SDS-PAGE and transferred (1 h 30 min at 390 mA) to a nitrocellulose membrane using a transfer apparatus according to the manufacturer’s protocols (Bio-Rad Laboratories Inc, Hercules, CA, USA). After incubation with 5% ECL blocking agent (GE Healthcare, Chicago, IL, USA) in PBST (Sigma-Aldrich, St. Louis, MO, USA), for 60 min, the membrane was washed once with TBST and incubated with antibodies against SB3 (anti-SerpinB3 mouse monoclonal antibody, clone 1C10, OriGene Technologies, Inc., Rockville, MD, USA, 1:1000 dilution), Haptoglobin (Abcam, Cambridge, MA, USA, 1: 1000 dilution), Fatty Acid Synthase (Cell Signaling, Danvers, MA, USA, 1: 1000 dilution) or β-Actin (Abcam, Cambridge, MA, USA, 1:400 dilution) at 4 °C overnight. Membranes were washed three times for 10 min and incubated with a horseradish peroxidase-conjugated anti-rabbit (Sigma-Aldrich, St. Louis, MO, USA, 1: 10,000 dilution) or anti-mouse (KPL, SeraCare Company, Gaithersburg, MD, USA, 1: 5000 dilution) antibodies for 2 h. Antigenic detection was carried out by an enhanced chemiluminescent substrate (Euroclone SpA, Milano, Italy) according to the manufacturer’s protocols and the densitometric analysis was assessed using the VersaDoc Imaging system (Bio-Rad Laboratories, Hercules, CA, USA).

### 3.5. Biological Activity of EV-SerpinB3 In Vitro 

#### 3.5.1. HepG2 Cells

A total of 40,000 cells/well of the HepG2 cell line were seeded the day before the experiment. The following treatments were then carried out: 250 μM H_2_O_2_ as a positive control, only the culture medium as a negative control, different concentrations of EVs from 1.0 × 10^7^ to 1.0 × 10^3^ EVs/mL (EVCTR and EVSB3) or recombinant SB3, obtained in our laboratory [17], at the same concentration of SB3 measured in EVs-SB3 in presence of 250 μM H_2_O_2_. All treatments were carried out for 72 h.

Cell survival was evaluated by MTT assay (Sigma-Aldrich, St. Louis, MO, USA) and the absorbance of the resulting purple solution was spectrophotometrically measured at 570 nm. The mortality rate induced by H_2_O_2_ was calculated vs. the untreated wells (CTR) and the protective effect of different EV preparations was then calculated vs. the H_2_O_2_ treated cells.

#### 3.5.2. Cardiomyocyte Primary Cells

Cardiomyocytes were seeded at 30,000 cells/well concentration the day before the assay.

The next day the same treatments carried out in the HepG2 cells were performed, but with an incubation time of 2 h, due to the higher sensitivity of the primary cells to H_2_O_2_ compared to the immortalized HepG2 cell line. The evaluation of cell survival was carried out, first using a Trypan Blue Staining Solution (Trypan blue 1:2, Sigma-Aldrich, St. Louis, MO, USA). The number of vital cells was measured with the Biorad TC20 tool™. The mortality rate induced by H_2_O_2_ was calculated vs. untreated (CTR) and the protective activity of EVs vs. H_2_O_2_ was also calculated.

A second assay was carried out to quickly discriminate the live cells from the dead cells by simultaneously staining with green-fluorescent calcein-AM to estimate intracellular esterase activity, and red-fluorescent ethidium homodimer-1 was used to indicate the loss of plasma membrane integrity (live and dead assay). The fluorescence was detected using a fluorescent microscopy Axiovert200 with Apotome2.1 (Zeiss, Oberkochen, Germany).

### 3.6. EV Labeling and Uptake In Vitro

A total of 5 × 10^10^ EVs in 1 mL of PBS and 1 mL of PBS alone (control) were labeled with the Vybrant™ DiI Cell-Labeling Solution (1,1′-dioctadecyl-3,3,3′3′-tetramethylindocarbocyanine perchlorate, Thermo Fisher, V22885) for 1 h at 4 °C, at 1 µM final concentration. Labelled EVs (EV-DiL) and PBS (PBS-DiL) were centrifuged at 4 °C at 2000 g with the 100 kDa filters (Amicon-Merck Millipore, UFC910096) for 15 min. Filters were washed two times to eliminate unbound dye. EV-DiL and PBS-DiL (150–200 mL) were retrieved. HepG2 cells were seeded on slides and the day after they were incubated with labelled EV preparations. The EV uptake was analyzed at 6 and 24 h after the start of treatment. The cells were fixed with paraformaldehyde and the nuclei were counterstained with DAPI.

The red staining of EVs internalized by the cells were detected and quantified using the Fluorescence microscopy Axiovert 200 uploaded with Apotome2.1 (Carl Zeiss MicroImaging GmbH, Göttingen, Germany).

## 4. Discussion

EVs are increasingly studied as vectors for drug delivery because they can transfer a variety of molecules across biological barriers. EVs can deliver bioactive molecules such as proteins and RNAs including siRNAs and microRNAs, with profound effects on the target cells. Compared with synthetic vectors, EVs exhibit better biocompatibility and can even show a particular tropism towards some tissues, either naturally or following surface modifications [18,19]. The transfer of cargo into EVs can be achieved either by exogenous loading, consisting in the incorporation of the therapeutic molecules into isolated EVs, or by endogenous loading, by genetically modifying the parental cell to overexpress a desired RNA or protein which is then secreted into EVs. In the present work, we adopted the latter method, which resulted in a high SerpinB3 concentration in the EVs secreted by the selected cell line. Indeed, exogenous loading by electroporation can injure the EV membrane, resulting in decreased biological activity, probably due to a decreased ability of the nanoparticles to fuse with the target cell membrane and deliver their cargo into the cytoplasm [20,21]. 

SerpinB3 has shown remarkable effects in protecting tissues and organs from oxidative stress [9] that could result also from ischemia/reperfusion injury. This effect was confirmed in our in vitro assays using both hepatoma cell lines and primary cardiomyocytes. Interestingly, the cytoprotective activity of Serpin B3 was strongly enhanced by encapsulation in the EVs. This phenomenon could result from an enhanced delivery of the molecule into the target cells, as suggested by the efficient cell uptake of the fluorescent labeled EVs, and/or from a more specific delivery to the intracellular site of action [19,20,22]. Alternatively, encapsulation into EVs could protect the protein from degradation. Additional studies are needed to verify whether this mechanism is present in vivo also. Importantly, the improved intracellular delivery of SerpinB3 could lead to a reduction in the required dose and of the risk of toxicity [20,23]. 

SerpinB3 expression was associated with relevant changes in the protein composition of secreted EVs that were mostly characterized by a reduced expression of other proteins compared with EVs from non-engineered cells. Indeed, it was estimated that each EV can accommodate a limited number of protein molecules, in a range of 10–30 [24,25]. It can thus be inferred that the secretion of SerpinB3 into the EVs will limit the loading of the other molecules. It is worth noting that the only protein overloaded in EV-SB3 compared to the control EVs was haptoglobin, an antioxidant molecule that has been found not only to modulate many aspects of the acute phase response but also to carry a pro-angiogenic potential [12]. The higher concentration of EVs detected in the conditioned medium of HepG2/SB3 cells, compared to the HepG2 control cells could be due to the increased Ca^2+^ concentration found in cells overexpressing SerpinB3 [26]. Indeed, Ca^2+^ affects both plasma membrane and cytosolic machinery with calpain activation, leading to plasma membrane EV biogenesis [27]. In addition, EVs are also considered as part of a process whereby cells get rid of undesirable proteins and molecules [28,29]. Therefore, it is plausible that cells will increment this pathway to eliminate the transgenic protein. 

The tissue-protective properties of EVs expressing SerpinB3 could be exploited for the protection and recovery from acute and chronic ischemic damage, as in patients with myocardial infarction or with critical ischemia of the lower limbs. These nanoparticles could also be used to improve the preservation of grafts before transplantation, especially those derived from non-heart-beating donors. 

A possible concern with the present approach in clinical applications is the use of a cancer cell line such as HepG2, raising warnings of possible tumorigenicity. However, tumor-derived EVs have been proposed and successfully tested as efficient drug carriers for chemotherapy in animal models [30]. Moreover, transgene expression could also be induced in primary cells such as mesenchymal stromal cells, whose secreted EVs exhibit intrinsic tissue-protective and pro-regenerative activities [1,31,32,33] that might be further improved by enrichment in SerpinB3. 

In conclusion, we have shown that inducing SerpinB3 transgene expression in HepG2 cell line results in the secretion of EVs enriched with the protein product, exhibiting enhanced cytoprotective activity compared to the nude protein and to naïve EVs. Additional in vivo studies are needed to evaluate if these engineered EVs can be used as therapeutic tools in diseases associated with ischemic injury and oxidative stress.

## 5. Patents

S.Q., A.B., M.R., P.P., M.M. are inventors of the Patent filed by the University of Padova N. 102019000002155, filing date 14 February 2019; PTC extension ongoing.

## Figures and Tables

**Figure 1 pharmaceuticals-14-00703-f001:**
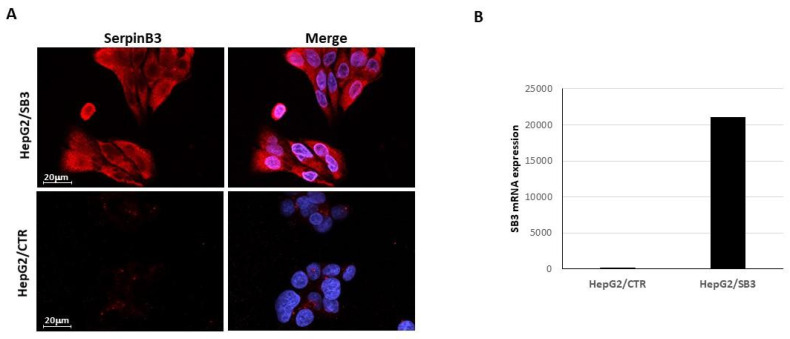
Example of SerpinB3 expression. (**A**) Representative immunofluorescence analysis of SerpinB3 expression in a SerpinB3 transfected HepG2 cell clone (HepG2/SB3) and in an empty vector transfected HepG2 cell clone (HepG2/CTR) used for EV preparation. The cells were immunostained with anti-SerpinB3 antibody (red) and the nuclei were counterstained with DAPI (blue). (**B**) Analysis of SerpinB3 (SB3) transcripts in the corresponding clones by quantitative real-time PCR (Q-PCR). The relative expression of SB3 mRNA is expressed by calculating 2-ΔCt. The experiment was performed in triplicate.

**Figure 2 pharmaceuticals-14-00703-f002:**
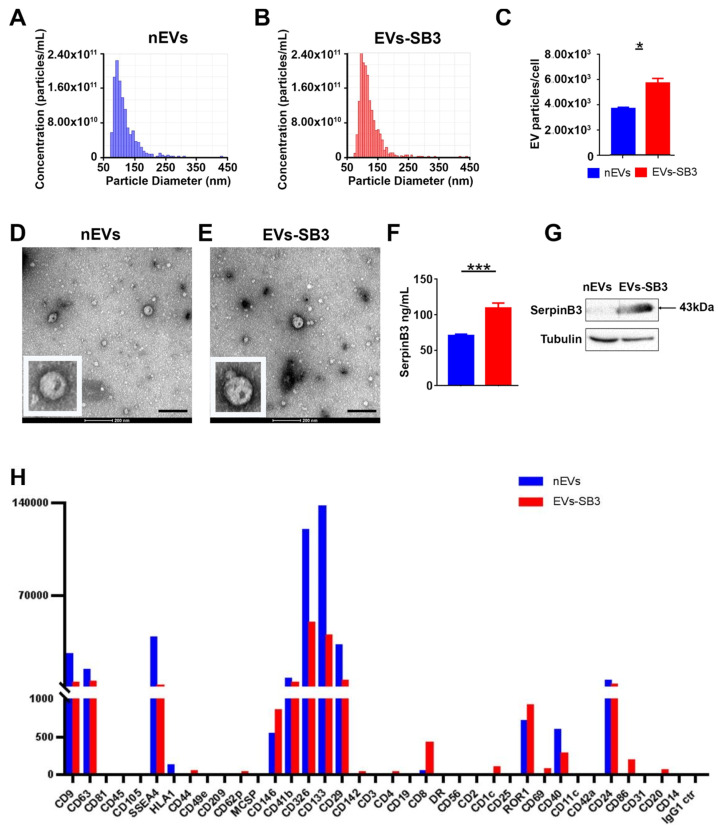
Characterization of extracellular vesicles derived from HepG2s. Representative size distribution of naïve EVs (nEVs) (**A**) and of EVs-SB3 (**B**) analysed by Resistive Pulse Sensing. (**C**) Number of nEVs and of EV-SB3 secreted per cell. Result are mean ± standard error (*n* = 5 independent experiments), * *p* < 0.05. Transmission electron microscopy analysis of freshly nEVs (**D**) and of freshly EVs-SB3 (**E**); scale bar refers to 200 nm. (**F**) SerpinB3 quantification by ELISA assay. Result are mean ± standard error (*n* = 5 independent experiments), *** *p* < 0.001. (**G**) Western Blot of nEVs and EVs-SB3 for SerpinB3. (**H**) Distribution of surface biomarker expression in nEVs and in EV-SB3.

**Figure 3 pharmaceuticals-14-00703-f003:**
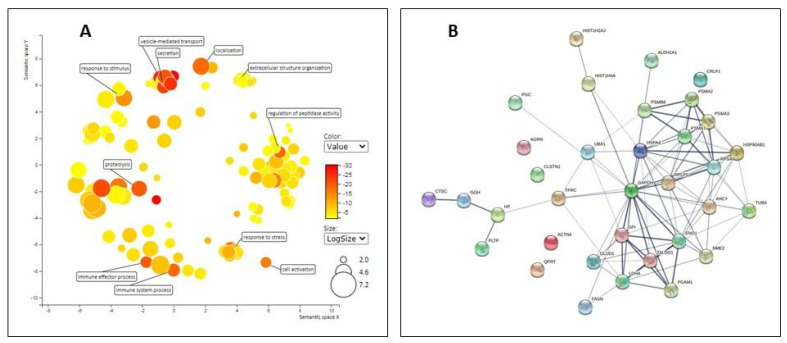
Bioinformatic analysis of proteins identified in EV preparations. (**A**) The enriched Gene Ontology (GO) terms that are associated to the proteins identified in EVs samples, as obtained by the gProfiler tool (p value > 0.01) and further processed with the Revigo software (*n* = 3 independent preparations). (**B**) STRING network of physical/functional interactions involving only the proteins that have a significantly different abundance in nEVs and in EVs–SB3.

**Figure 4 pharmaceuticals-14-00703-f004:**
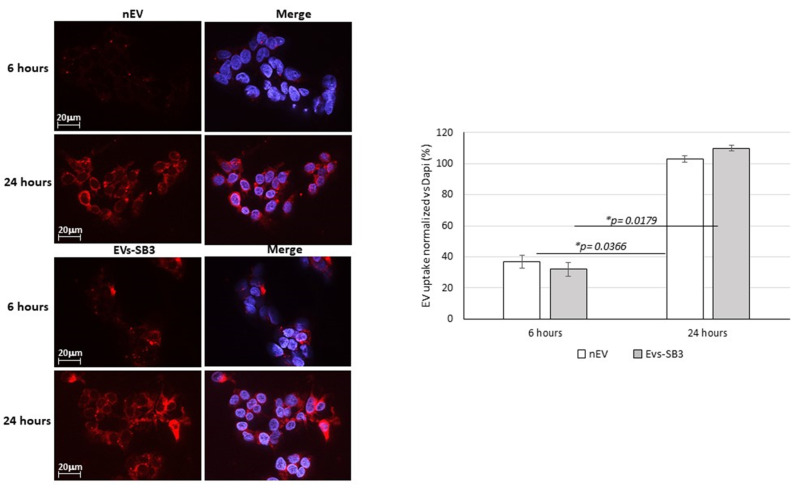
Left panel: Fluorescence expression of DIL-EVs uptake in HepG2 cells over a 6–24 h timeframe. Original magnification 63×. Right panel: Graphic representation of DIL-EVs quantification in HepG2 cells. The negative controls were subtracted from the fluorescence intensity of nEVs and EVs-SB3 and final results were normalized against staining of the nuclei with Dapi. The experiment was performed in triplicate.

**Figure 5 pharmaceuticals-14-00703-f005:**
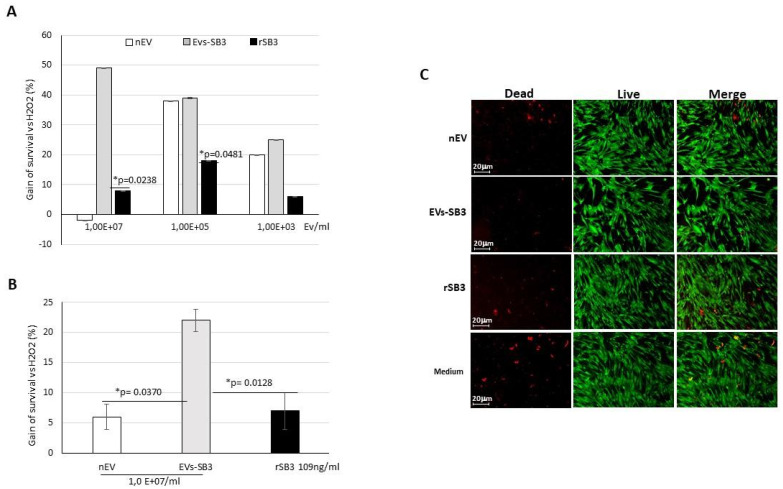
Protective activity of different EVs preparation and of recombinant SerpinB3. (**A**) Histograms represent the percent of gain of survival of HepG2 cells treated with 250 μM H_2_O_2_ for 72 h in presence of different EVs or of recombinant SerpinB3, compared to untreated cells. SB3 concentration was identical to that detected in the corresponding EVs-SB3 preparations. (**B**) Histograms represent the percent of gain survival in cardiomyocytes treated with 250 μM H_2_O_2_ in presence of different EVs preparations at 1.00 × 10^7^/mL concentration or of recombinant SerpinB3, compared to untreated cells. Both experiments were performed in triplicate. (**C**) Fluorescence images of Live and Dead assay in primary cardiomyocyte treated for 2 h with 250 μM H_2_O_2_ in presence of different EVs preparations or of recombinant SerpinB3. Original magnification 63×.

**Table 1 pharmaceuticals-14-00703-t001:** Differently abundant proteins detected in nEVs and in EVs-SB3 preparations.

Gene Name	# Unique Peptides	Fold Change (SB3 vs. CTR)	*p* Value
HP	17	2.1	8.6 × 10^4^
GGH	4	−2.2	1.9 × 10^2^
ACTN4	17	−2.3	3.3 × 10^5^
TUBA1B	3	−2.4	3.4 × 10^2^
PSMA3	2	−2.6	3.9 × 10^2^
ENO1	17	−2.7	4.6 × 10^2^
TALDO1	2	−2.8	5.0 × 10^2^
LDHA	8	−3.1	1.8 × 10^2^
NME2	8	−3.2	1.2 × 10^3^
CLSTN1	13	−3.2	6.5 × 10^3^
HSP90AB1	18	−3.4	4.8 × 10^2^
QPRT	4	−3.5	3.5 × 10^2^
PSMB6	3	−3.7	1.3 × 10^2^
PSMA7	4	−3.7	4.1 × 10^2^
GAPDH	14	−3.7	4.9 × 10^2^
GPI	5	−3.7	0.0 × 10^0^
HIST1H2AJ	5	−3.7	8.0 × 10^4^
HSPA4	8	−3.8	2.8 × 10^4^
ALDH1A1	11	−4.0	0.0 × 10^0^
RPSA	4	−4.1	1.5 × 10^2^
CTSC	2	−4.2	4.1 × 10^8^
PGAM1	2	−4.2	2.9 × 10^6^
AHCY	9	−4.3	5.0 × 10^3^
UBA1	5	−4.3	2.2 × 10^2^
FASN	44	−5.0	1.6 × 10^2^
PSMA2	2	−5.8	1.2 × 10^2^
GLUD1	15	−6.4	2.2 × 10^3^
HIST1H4A	7	−7.2	1.8 × 10^2^
TFRC	5	−14.6	1.4 × 10^7^
AGRN	10	−20.2	8.8 × 10^6^
ACLY	3	Found only in CTR	---
APEH	2	Found only in CTR	---
VCAN	7	Found only in CTR	---

## Data Availability

Data supporting reported results can be found in laboratory databases of the authors.

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
