# Peer review of "Engineered EVs for Oxidative Stress Protection"

_pharmaceuticals, 2021, doi:10.3390/ph14080703_

Round 1
Reviewer 1 Report
The manuscript "Engineered EVs for Oxidative Stress Protection" by Tolomeo et al. investigates the protective role of SerpinB3 overexpressed on the surface of EVs derived from engineered HepG2.
The authors conducted rigorous experiments to support their results, but in my opinion there are some points to be clarified.
Major points:
- In the introduction, I suggest to explain what type of EVs were the object of the study, small or medium/large EVs. I would suggest that the authors review: Minimum Information for Extracellular Vesicle Studies 2018 (MISEV2018): a position statement from the International Society for Extracellular Vesicles and an update of the MISEV2014 guidelines; Journal of Extracellular Vesicles 2018, Vol 7, 1535750.
- Can the authors explain the results obtained in Figure 2C? Why, in their opinion, the EVs concentration, expressed in particle/cell, increased twice in the conditioned medium of HepG2 SerpinB3 compared to the naive EV?
Minor point
- line 119 there is probably a typo error "0.22mm"
- line 155 there is probably an error "50 g of protein"
- line 203, 207, 317 and 328 there is probably an error in the H2O2 concentration
Author Response
The manuscript "Engineered EVs for Oxidative Stress Protection" by Tolomeo et al. investigates the protective role of SerpinB3 overexpressed on the surface of EVs derived from engineered HepG2.
The authors conducted rigorous experiments to support their results, but in my opinion there are some points to be clarified.
Major points:
- In the introduction, I suggest to explain what type of EVs were the object of the study, small or medium/large EVs. I would suggest that the authors review: Minimum Information for Extracellular Vesicle Studies 2018 (MISEV2018): a position statement from the International Society for Extracellular Vesicles and an update of the MISEV2014 guidelines; Journal of Extracellular Vesicles 2018, Vol 7, 1535750.
We agree with the Reviewer that a more accurate description of the EV subtype is desirable, following MISEV2018 guidelines, and we apologize for the inaccuracy. As described in the Methods section, culture medium was passed through a 0,22 um filter. Both RPS and TEM analysis confirmed that almost all nanoparticles were below 200 nm in size. Therefore, the nanoparticles isolated in the present work can be referred to as “small EVs” according to the MISEV2018 criteria (Théry et al. 2018). This information was added to the text (page 10 lines 198-200) and in new ref. 16.
- Can the authors explain the results obtained in Figure 2C? Why, in their opinion, the EVs concentration, expressed in particle/cell, increased twice in the conditioned medium of HepG2 SerpinB3 compared to the naive EV?
The increased concentration in the conditioned medium of HepG2/SerpinB3 compared to the naive EV could be due the increased Ca2+ concentration, found in cells overexpressing SerpinB3. Ca2+ indeed affects both plasma membrane and cytosolic machinery, with calpain activation, leading to plasma membrane EV biogenesis. In addition, EVs are also considered a process whereby cells get rid of undesirable proteins and molecules. Therefore, it is plausible that cells will increment this pathway to eliminate the transgenic protein. These considerations were added to the text (pages 13-14, lines 271-276) and in new ref. 26-29.
We are grateful to the reviewer for addressing the attention to minor points that were corrected, as suggested.
Reviewer 2 Report
Manuscript presented by Tolomeo and co-workers entitled “Engineered EVs for Oxidative Stress Protection” described the protection effects of extracellular vesicles overexpress SerpinB3 (EVs-SB3) on oxidative stress-induced damage and cytoprotective activity. In my opinion the paper needs major revision, other experiments are needed, more data should be presented.
Major: Only in-vitro experiments were shown, In vivo experiments are required to confirm the cardioprotection of EVs-SB3.
My minor comments are presented below. Provide the explanation for all of them, make changes in the text.
Why did you choose HepG2 cells for the production of EVs?
Figure 1A, please add a scale bar and mean and SD for Figure 1B, is performed in triplicate?
Figure 4, please add scale bar. In p value add a dot for decimals, not a comma.
Figure 5C, please add scale bar.
Please mention many times experiments were performed (triplicate or more?) in legends of all figures, wherever applicable.
How did you decide the concertation of 109 ng/mL for rSB3?
In results – you said SerpinB3 was detected only in EVs-SB3, as demonstrated by Elisa and Western blot results but results in figure 2F showed SerpinB3 was around 60-70 ng/ml and western also showing faint band.
Author Response
Manuscript presented by Tolomeo and co-workers entitled “Engineered EVs for Oxidative Stress Protection” described the protection effects of extracellular vesicles overexpress SerpinB3 (EVs-SB3) on oxidative stress-induced damage and cytoprotective activity. In my opinion the paper needs major revision, other experiments are needed, more data should be presented.
Major: Only in-vitro experiments were shown, In vivo experiments are required to confirm the cardioprotection of EVs-SB3.
We agree with the Reviewer that in vivo experiments are required to confirm therapeutic efficacy, and we have submitted to our National Ethics Committee a project to test these nanoparticles as a tool to improve graft function for organ transplantation in a large animal model. However, we still believe that the present data, demonstrating the feasibility of the procedure, the effects on EV cargo composition and an in vitro proof-of-principle of efficacy could be of interest to the scientific community.
My minor comments are presented below. Provide the explanation for all of them, make changes in the text.
Why did you choose HepG2 cells for the production of EVs?
Human hepatocarcinoma cell lines (specifically, Bel7402 cell line) have already been used to produce EVs for drug delivery, as reported in ref. 25 (Nat Commun, 2019, 10, 1–16). We thus adopted a cell line of analogous origin to develop the transfection procedure and to obtain in vitro evidence of efficacy. Clearly, following your comment, different cell lines could be used – or even primary cells, as stated in the discussion.
Figure 1A, please add a scale bar and mean and SD for Figure 1B, is performed in triplicate? Figure 4, please add scale bar. In p value add a dot for decimals, not a comma. Figure 5C, please add scale bar. Please mention many times experiments were performed (triplicate or more?) in legends of all figures, wherever applicable.
The requested details have been added in the revised figures 1,4,5 and the information that the experiments were performed in triplicate was added to the legends of the figures.
How did you decide the concertation of 109 ng/mL for rSB3?
The concertation of 109 ng/mL was the concentration of SerpinB3 detected by ELISA in the EVs used in the reported experiment and therefore, to assess the effect of the protein alone, we have used this identical concentration of SerpinB3.
In results – you said SerpinB3 was detected only in EVs-SB3, as demonstrated by Elisa and Western blot results but results in figure 2F showed SerpinB3 was around 60-70 ng/ml and western also showing faint band.
We agree with the reviewer that a faint band is visible in Western blot ant therefore we acknowledge your suggestion to modify the results description, reporting that naïve EV showed trivial levels of SerpinB3. It should be noticed that SerpinB3 concentration, detected in naïve EVs by ELISA were around the detection limit of the assay. This information has been added to the text (page 10, lines 200-201).
Round 2
Reviewer 2 Report
Authors have answered my most of questions and concerns even though in vivo experiments were not performed at the moment.